# Neoplastic Implications in Patients Suffering from Hidradenitis Suppurativa under Systemic Treatments

**DOI:** 10.3390/biomedicines9111594

**Published:** 2021-11-01

**Authors:** Federica Li Pomi, Laura Macca, Alfonso Motolese, Ylenia Ingrasciotta, Massimiliano Berretta, Claudio Guarneri

**Affiliations:** 1Department of Clinical and Experimental Medicine, Section of Dermatology, University of Messina, 1-98125 Messina, Italy; federicalipomi@hotmail.it (F.L.P.); lauramacca7@gmail.com (L.M.); alfonsomotolese93@gmail.com (A.M.); 2Department of Biomedical and Dental Sciences and Morphofunctional Imaging, Section of Pharmacology, University of Messina, 1-98125 Messina, Italy; yingrasciotta@unime.it; 3Department of Biomedical and Dental Sciences and Morphofunctional Imaging, Section of Infectious Diseases, University of Messina, 1-98125 Messina, Italy; berrettama@gmail.com; 4Department of Biomedical and Dental Sciences and Morphofunctional Imaging, Section of Dermatology, University of Messina, 1-98125 Messina, Italy

**Keywords:** hidradenitis suppurativa, cancer, cutaneous tumours, noncutaneous tumours, haematological malignancies, HS treatment

## Abstract

Hidradenitis suppurativa (HS) is a chronic, recurrent, inflammatory skin disease of the apocrine glands. It typically involves the axillary, submammary, genital, inguinal, perineal, and perianal regions. The development of abscesses, sinus tracts, and scars can lead to pain, scarring, disfigurement and decreased quality of life. HS is associated with a wide range of comorbidities. Several studies of co-occurrence of HS and nonmelanoma skin cancer suggest a causal relationship. In an attempt to assess the link between HS and cancer, we performed a systematic review of the current scientific knowledge through a PubMed-based literature search. Results show that HS could be associated with an overall risk of cancer and numerous specific cancers such as: nonmelanoma skin cancer (NMSC), hematologic malignancies, and metastatic cancer. Among NMSC, squamous cell carcinoma (SCC) is considered the most common complication arising in long-standing HS. Based on our review, we suggest that cautious surveillance and active intervention may be warranted in patients with HS. Moreover, an age-appropriate cancer screening should be offered to all patients, especially those who developed HS later in their life or in long-standing moderate to severe HS with multiple comorbidities.

## 1. Introduction

Hidradenitis suppurativa (HS), also known as acne inversa or Verneuil’s disease, is a chronic, recurrent, inflammatory skin disease of the apocrine glands. It typically involves the axillary, submammary, genital, inguinal, perineal, and perianal regions, leading to the development of abscesses, sinus tracts and scars [1,2]. Though it is often misdiagnosed, HS is a relatively common condition, with a prevalence of up to 4%, mainly affecting women in their third and fourth decades of life [3]. Prevalence, indeed, is greater among women (75%) than among men (25%). HS begins with occlusion of a hair follicle, followed by hyperkeratosis, hyperplasia, and dilatation. The rupture of the follicle leads to the release of keratin and bacteria into the dermis [4]. This disruption produces a significant inflammatory response, recruiting self-perpetuating inflammatory mediators [5]. There is a considerable overlap between risk factors and comorbidities [6,7]. A sizable body of literature demonstrates obesity as a paramount risk factor; indeed, the adipose cells are considered an independent endocrine tissue capable of secreting proinflammatory cytokines, which may add to the chronic inflammatory state of HS [6]. The association with smoking was also related to clinical HS disease severity. It was recently proposed that nonsmoking and nonobesity is associated with a better chance of HS remission, and it has come to light that HS surgery combined with smoking cessation give rise to fewer or no lesions [6]. There is also an association with autoimmune comorbidities with a significant overlap between patients with HS and inflammatory bowel disease (IBD), especially Crohn’s disease (CD) [6,8]. Several case reports of co-occurrence of HS and nonmelanoma skin cancer suggested a causal relationship, but firm epidemiological data are lacking [9,10]. In an attempt to assess the link between HS and cancer, we performed a systematic review of the current literature on the topic.

## 2. Materials and Methods

We checked the PubMed (https://ncbi.nlm.nih.gov/PubMed) and EMBASE databases using the string “Hidradenitis suppurativa” [All Fields] AND “Cancer” [All Fields], without time limits. The systematic literature search was led according to PRISMA flowchart, also reviewing the abstracts of the articles whose title suggested this association. Only papers written in English language and concerning humans were included. The references retrieved were critically examined by two experts in the field of dermatology to select those pertinent, thus reporting the type of the selected articles (review, retrospective cohort study, case control study, case series, case report and literature review, case reports, letter to editor, commentaries). Based on the type of study, the strength of evidence for each article was graded (Table 1). The reference lists of these papers were also examined to find other relevant articles, which were eventually revised and included if appropriate. The risk of bias of the observational studies included in this systematic review was independently assessed by two authors (CG and LM) using the Newcastle-Ottawa quality assessment scale [11]. This instrument consists of eight different domains for cohort studies (representativeness of the exposed cohort, selection of the non-exposed cohort, ascertainment of exposure, demonstration that outcome of interest was not present at start of study, comparability of cohorts on the basis of the design or analysis, assessment of outcome, follow-up long enough for outcomes to occur, adequacy of follow up) and case-control studies (adequate case definition, representativeness of the cases, selection of controls, definition of controls, comparability of cases and controls on the basis of the design or analysis, ascertainment of exposure, same method of ascertainment for cases and controls, nonresponse rate). The included studies were categorized as “low risk of bias” if at least six of the eight domains were judged to be at low risk of bias. The overall risk of bias was rated as low for 6 pertinent studies [9,12,13,14,15,16] as the rest were represented by reviews (12), case reports (41), case series (7), and commentaries (3).

## 3. Results

As of 23 October 2020, a PubMed search of “Hidradenitis suppurativa” AND “Cancer” yielded 276 articles of which 207 were not considered because the title and/or abstract suggested that they were not concerning topics, not written in English, not performed on human populations only, not relevant to the outcome of interest, and/or because the full text was not available (Figure 1). The other 69 concerned the association between HS and cutaneous tumours (*n* = 64), noncutaneous tumours (*n* = 1), and haematological malignancies (*n* = 4). Among the 64 articles on cutaneous cancer, we selected 12 reviews, 2 retrospective cohort study, a case control study, 6 case series, 6 case reports and literature review, 31 case reports, 3 letter to editor, and 3 commentaries. The only article about noncutaneous tumors was represented by a retrospective cohort study. With regard to the association with hematological malignancies, we found a cross sectional cohort study, together with a case control study, a case series and a case report. First author, year of publication, type of study, study population, and strength level of each article are summarized in Table 2.

## 4. Discussion

### 4.1. HS and NMSC

Many studies found an increased risk of NMSC among patients with HS [65], with a prevalence of about 4.6% [4]. In particular, the risk of SCC development in HS seems to be overall rare and it was estimated to be between 0.5% and 5.2% [4,12,38,41,46,51,66,67,68,69,70,71,72,73,74]. The pathogenesis of this phenomenon is still debated and Fabbrocini et al., through a meticulous literature research starting from the 1950s and identifying over 90 cases of HS patients developing SCC mostly on perineal or buttock areas, theorized on an ‘immunocompromized cutaneous district’ to explain its occurrence [55] on a chronically diseased skin district with a locally dysfunctional immune control. The ‘loss’ of immune control, therefore, allows the development of a dysimmune reaction, an infection, or a tumour. This represents a novel concept that applies to some HS lesional areas and in which many additional variables were called into question. Some authors speculated on the link between HS and the development (and progression) of cancer from a pathogenetic point of view. A study of Kurokawa focused on changes of cytokeratin (CK) expression in two cases of well-differentiated and poorly differentiated SCC arising from HS lesions. In both cases, type A (infundibular-like keratinized) epithelia were observed. In type A epithelia, CK 1 and 10 expressions were decreased, and CK 14 and 17 were detectable in the whole layers. In tumor nests of well-differentiated SCC, CK 1, and 10 expressions were downregulated, and CK 14 expression was upregulated. In tumor nests of poorly differentiated SCC, CK 1 and 10 were not expressed, but simple epithelial keratins (CK 8, 18 and 19) were expressed. These changes of CK expression are related to malignant transformation from the type A epithelium in HS to SCC and represent a new concept that attempts to explain the pathogenesis of this disease [35]. Lastly, another reading key about the pathogenic mechanism for the SCC development in long-standing HS, may well be explained by the loco-regional immune default. This concept refers to a skin site of locoregional immune dysregulation. The cause could be due to an obstacle to the normal trafficking of immunocompetent cells through lymphatic channels and/or interference with the signals sent by neuropeptides and neurotransmitters, which are related to peripheral nerves and to cell membrane receptors of immunocompetent cells. Disruption of lymph microcirculation and damage to peripheral nerve endings can obviously occur in scarred skin and alter the local interplay between immune cells conveyed by lymph vessels and neuromediators running along peripheral nerve fibers. Depending on which of the neurotransmitters and immune cells are involved, this destabilization may be either defective, thus predisposing to infections and tumours (such as SCC), or excessive, thus favouring the occurrence of some immune disorders or dysimmune reactions at the sites “marked” by scarring. Other potential risk factors may contribute in (and possibly bias the rates of) this malignant transformation, including cigarette smoking [4,6,75], which is a common finding in 80 to 90 per cent of HS patents, and HPV infection. The first was hypothezised to create a proinflammatory environment through the release of cytokines such as TNF-alpha. On the other hand, oncogenic HPV16 infection was demonstrated in severe hidradenitis suppurativa (HS) of the anal, perianal, gluteal, thigh, and groin regions evolving into squamous cell carcinoma (SCC) [76]. According with Schenfield et al., the development of malignancies in these sites may occur owing to a synergistic effect between the chronic inflammation of HS, impaired cellular immunity and the presence of the HPV [43]. Furthermore, HPV is well known to play an important role in the pathogenesis of mucosal and skin SCC as it prevents apoptosis, allowing continuous viral DNA replication. A review of literature by Jourabchi et al. demonstrated that, among patients with long-standing HS that developed SCC, 72.7% were positive for alpha-Human Papilloma Virus (HPV) and 87.5% were positive for beta-HPV [57]. Other studies by Lavogiez et al. [41] and Flores et al. [77] confirmed a connection between HPV-16 infection and SCC. Finally, the association between HPV and HS remains unclear [13]. In some cases, despite the histological evidence of HPV, (represented by koilocytotic atypia), the polymerase chain reaction (PCR) remains uncertain [30]. Thus, HPV may act as a local cofactor, alone or in assocition to others, on a substrate of chronic inflammation to cause cutaneous SCC in these areas [30]. In this setting, the newly approved nonavalent HPV vaccines have demonstrated safety and efficacy against HPV oncogenic strains and allow to suppose a potential protective, and even promising curative effect, in these patient subsets. The risk of acquiring HPV resulted also heightened by using some biotechnological drugs in HS management. As these treatments work by attenuating the host immune response to reduce inflammation, on the other side they also leave the patient at risk for reactivating latent infections [41,78,79]. In fact, additional concerns regarding the development of SCC in the course of biological treatment have arisen [80,81]. In some patients with HS the use of TNF antagonists, like adalimumab, increases up to a two-fold the risk of developing SCC and may shorten the time to the development of NMSC, particularly in psoriatic patients [60,76,82,83]. However, in 2016, Verdelli et al. [47] described a fatal case of sepsis and SCC complicating HS after treatment with infliximab for severe and long-standing HS. The possible role that the TNF-alpha inhibitor played in the development of SCC on HS lesions, and concomitant sepsis, was unclear, because there was just a significant baseline immune dysregulation in their patient and an increased infectious risk were clearly reported with this drug class [6]. How and to what extent TNF-alpha inhibitors may promote the development of NMSC still remains controversial [10]. Sure enough, a meta-analysis of 74 randomized controlled trials failed to confirm or refute that anti-TNF-alpha agents caused an increase in the development of malignancy [84]. To the latest research, no direct causal relationship between the use of TNF-alpha or other biological agents (e.g., anti-IL 12/23, anti IL-17, and anti IL-23, also anecdotically used in HS patients) and tumours development were reported. Notwithstanding the above mentioned pathogenic hypothesis [55], that makes these patients prone to turn some of their HS lesions in to the development of a tumour or an infection, the administration of the so called ‘biologics’ seem to not provide any additional risk. Among cutaneous tumors, the occurrence of Marjolin ulcer (MU) was reported in several studies. It is a rare and aggressive malignant degeneration to SCC characterized by scars, long-standing wounds, ulcers, or chronic inflammation with an high incidence in the setting of HS. The 2.78% incidence rate of MU described by Yon et al. in their study on HS patients is comparable to the historical incidence of MU in burns (2%) [53]. The exact prevalence of MU arising from HS lesions cannot be determined until more prospective data can be obtained, but some authors, such as Huang et al., estimate it as up to 4.1% [51]. Athough HS is more prevalent in females, MU occurring in HS patients demonstrates a 6.75:1 male/female ratio [51]. The etiology of this malignant transformation is unknown, but some phisicians postulate that chronic irritation, ulceration and tissue repair lead to a weakened epithelium that is more susceptible to carcinogens [62]. Strict criteria were proposed to classify a carcinoma as a MU. Proof that trauma contributed to the development of cutaneous malignancy requires that: (a) the skin site was normal before injury, (b) the injury was severe enough to disrupt tissue continuity, (c) the malignancy arose in the site of injury and (d) the malignancy was the result of attempted regeneration and repair of injured tissues [44]. MU is rare, but it is a life-threatening complication of HS, due to its increased metastatic potential and high lethality. High sensitivity in suspicion and prompt management of cutaneous SCCs is fundmental to avoid this complication. Chronic HS lesions, especially in the gluteal region, should be carefully observed: cautious surveillance and active intervention should be taken especially in the ulcerating type ones [51]. Once tumor has arisen from HS lesion, immediate radical excision should be performed [51]. With assured clear margin, topical negative pressure (TNP) could be chosen to offer a favorable environment for the survival of skin grafting [51]. Indeed, several cases in literature presenting with long-standing HS which developed into SCC are reported [12,38,39,41,46,51,66,67,68,69,70,71,72,73,74]. There are several difficulties in the diagnosis of SCC in the context of chronic HS lesions, because the presentation of the tumor as a nodule or ulcer may clinically mimic classical HS presentation. Therefore, patients with HS should be followed up at least once every 6 months and assessed for new persistent or rapidly growing lesions, even in the context of long-standing disease [39]. Early recognition of these lesions is essential as a delay in diagnosis can convert a potentially curable lesion into an incurable one. To that end, it is of utmost importance that the physician takes special note of any chronic and longstanding wound since these seem to have a higher likelihood of malignant potential [39]. Unfortunately, despite this widely acknowledged concept, it is still possible to overlook and underdiagnose a malignancy concealed within an indolent lesion [39,85]. Up to date, there are no standardized monitoring guidelines for patients with HS regarding prevention of SCC. Ward et al. recommend a vigilant approach to HS patients with chronic ulcers and/or verrucous changes, whereas scouting biopsies should be performed to rule out SCC [64]. If biopsies are negative, the authors propose obtaining a minimum of HPV 16 and 18 testing via immunostaining of biopsied skin or other cytologic sampling [64]. In addition, if there is evidence of verrucous changes on examination or on histology despite negative HPV results, they encourage prophylactic treatment to decrease the risk of malignant conversion with possible use of interferon alpha-2b, 18 million unit/mL (1 mL) subcutaneous injections three times per week for four weeks [64]. After treatment, the patient should continue close clinical skin evaluation [64]. SCC has resulted to be most common in men with gluteal [17,21,22,23,25,39,42,54,86], perianal [18,19,27,40], and perineal [20,42,58] disease many years after the initial onset of HS, with negligible axillary cases yet reported [13,59]. A possible explanation for the anatomical predilection and male predominance for malignant transformation is that males with HS tend to have high morbidity in the anogenital and perineal regions, whereas females tend to have mainly the involvement of the axillary and inframammary regions [53]. The presence of sinus tracts in HS provides an easy route for dissemination of malignant cells, and the detection of malignant transformation may be difficult on the background of chronic tissue inflammation [53]. Furthermore, cases of SCC arising at the vulvar region [14,29,33,48], as well as perianal and perineal in women with a long history of the disease are not rare; difficulties in recognition may lead to delay in diagnosis [68]. The literature suggests that the local invasive recurrence rate for primary tumors (T1) is 7.2% after radical local excision compared with 6.3% after radical vulvectomy. Surgical margins must be at least 1 cm. Local recurrences usually can be treated successfully if diagnosed early, since recurrence in the groin is usually fatal [14]. Many authors have advocated that HS arising in extra-axillary sites be regarded as a pre-malignant condition and not to be treated conservatively. Close follow-up and repeated skin biopsies should be performed in those with suspected malignancy [48]. The review of pertinent literature also revealed anecdotical cases of tumours other than SCC arisen on HS lesions or SCCs developing in the setting of syndromic conditions. Alkeraye et al. described a case of a 61-year-old man with a history of chronic HS complicated by the development of mucinous adenocarcinoma, highlighting the importance of early detection and treatment for HS [28]. No signs of cancer were observed after abdominoperineal resection, which was followed by adjuvant radiation and chemotherapy [28]. However, the association between mucinous adenocarcinoma and HS is poorly reported being due to the lack of reporting or simply to the rarity of cancer development itself in the chronic course of HS [28]. SCC may also associate with HS in the context of Dowling Degos (DD) syndrome, because of the possible common genetic background and/or the fact that the two conditions share a similar follicular-based pathogenesis, represnted by repetitive inflammation and chronic scarring secondary to follicular/apocrine gland occlusion [87]. Furthermore, a case with a history of recalcitrant HS of the groins in the course of Keratitis-Ichthyosis-Deafness (KID) syndrome was described and then treated with *en bloc* resection, which lead to the discovery of a moderately differentiated SCC upon histological examination [34]. Although cutaneous SCCs are usually well differentiated, they could have a more aggressive behaviour with local invasion and distant metastasis. As we stated above, HS arising in extra-axillary sites represents a pre-malignant condition and should be treated with wide and radical surgical excision due to the risk of developing aggressive SCCs [45]. Many authors describe fatal cases of SCC arising in severe long-standing HS of the perineal and perianal sites or of the buttocks, often complicated by inguinal lymph node metastases [35]. Joglekar et al. reported a case of a 46-year-old male with a history of MU secondary to gluteal HS who developed bilateral pleural effusions, pleural thickening and scattered nodular densities within both lungs concerning for metastatic disease; the wide local excision of sacro-gluteal SCC occurred too late in this case, as the patient already had metastatic disease to inguinal lymph nodes before surgery [50]. SCC of the perineal area metastatising to the inguinal lymph nodes and to the liver metastases was also reported [26]. Similarly, two invasive SCCs of the vulva, developing in the course of HS, gave metastases repectively to the locoregional lymph nodes [52] and gland metastasis under the diaphragm [63] with fatal outcome in three and one months after the diagnosis. Additionally, Yen et al. described a fatal case of SCC of the buttock in a 45-year-old man affected by HS of the buttocks, complicated by metastasis to diffuse areas over the posterior neck, chest wall, abdominal wall, liver, bilateral thighs and pelvic floor [61]. The same body site was involved in another case description by Perez. Diaz et al. whereas primary tumour and locoregional lymph node metatstasis were treated with a favorable outcome at one-year follow-up [24]. Barresi V. et al. in 2008 firstly reported the occurrence of diffuse malignant peritoneal mesothelioma (DMPM) in a 47-year-old man affected by HS. The patient originally presented to our University hospital with ascites and abdominal pain: paracentesis and cytologic examination of the ascitic fluid were performed, and in light of the cytologic findings, a peritoneal biopsy was also carried out. The patient died 3 months after the diagnosis of DMPM was made. Clinical history was negative for exposure to asbestos nor chronic irritation or irradiation of the peritoneum, but it was significant for recurrent perianal HS of more than 25 years duration. As a sure causal link between the two conditions had been disputed, the clinical course of events seemed and seems more than coincidental [37]. Moreover, a case of SCC arisen in extensive perineal HS was associated with paraneoplastic neuropathy in a 50-year-old-man. The patient presented with subacute muscle weakness and sensory symptoms before the diagnosis of carcinoma was made by the consultant dermatologist, and completely resolved after the excision of the tumor [32]. A case of suspect paraneoplastic syndrome was finally described by Chang et al. who reported of a patient with large, open, malodorous wounds of the buttocks being infected by Pseudomonas, and abdominal Computed tomography (CT) showing pancreatitis and deep venous thrombosis in the inferior vena cava and bilateral femoral veins. Multiple biopsies taken from skin lesional area resulted positive for aggressive SCC. On the other hand, autopsy revealed severe hemorrhagic pancreatitis, pulmonary edema and disseminated SCC [44]. As nodal metastasis represents the most important prognostic factor for patients with SCC, late diagnosis of such tumours in nonaxillary HS is a great concern, thus resulting in extremely poor outcomes in these patients. According with a Medline search by Mac Leon et al., the two-year survival after the diagnosis was reported at only 52% rate [36]. By the way, the role of postoperative chemotherapy is uncertain and controversial but should be considered in case of inoperability and metastasis; radiation therapy alone or in the presence of surgery is only mildly effective at treating metastatic disease [36]. In summary, if HS is theoretically a benign entity, the high recurrence rate and the above mentioned subsequent clinical events makes it a really insidious condition. Lowering these rates together with prevention of complications represent the main target for the physician, that can be achieved only by timely medical intervention and elective monitoring of cases of longstanding, mainly nonaxillary, moderate to severe forms. Deep tissue biopsies are recommended and, if malignancy is suspected, a radical wide resection of all apocrine glands bearing cutis in the affected area is mandatory. Conservative surgical approaches lead to an unacceptable rate of recurrence with the risk of subsequent SCC [15,31,63]. Long-term local and/or systemic treatment after surgery with monitoring to exclude late recurrences is the futher step in the management of cancer risk.

### 4.2. HS and Tumors beyond the Skin

The connection between HS and overall cancer was also suggested: a retrospective study of 2119 patients conducted by Lapins et al. reviewed literature and assessed the prevalence of cancer in patients with HS, thus founding a 50% increased risk of all types of neoplasms compared with the standard rate in Swedish population [9]. In particular, apart from nonmelanoma skin cancer, buccal and liver cancers were significantly higher among HS patients. They also found that, although not statistically significant, the incidence of esophageal, lung, kidney, urinary tract and hematopoietic cancers increased among HS patients in comparison with that of general population. In part, the increased presence of malignancy was linked with the higher consumption of alcohol and smoking habits among HS patients [9]. However, the participants of this study might have had more comorbidities, which could increase the risk of cancer, than those observed in a nationwide, population-based study conducted by Jung et al. in the Republic of Korea [88]. In the latter, after adjustment of the hazard ratio (HR) for comorbidities, the overall risk of cancer was significantly higher in patients with HS than in the control patients, especially in those affected by moderate to severe forms [88]. Interestingly, also in their research the authors recognized a significant higher risk for oral cavity and pharyngeal cancer, together with some cases of Hodgkin lymphoma and, in order, central nervous system neoplasms, NMSC, the prostate and colorectal cancers [88]. In conclusion, they suggested that more intense cancer surveillance may be warranted in patients with HS and that lifestyle modifications in patients with HS to address behaviors such as smoking or alcohol use or conditions such as obesity should be emphasized.

### 4.3. HS and Hematological Malignancies 

The development of HS later in fifth-sixth decade may be associated with hematologic cancers. This is a particular cluster of HS patients, firstly observed by Sotoodian et al. in their small retrospective analysis. The study reported of 2 patients with a long-standing history of hematopoietic cancers, a 60-year-old male patient with hairy cell leukemia, and a 68-year-old male patient with chronic lymphocytic leukemia, who received no continuous treatments for their malignancies and developed late-onset HS [56]. In a cross-sectional cohort analysis Tannenbaum et al. investigated prevalence of non-Hodgkin lymphoma (NHL), HL, and cutaneous T-cell lymphoma (CTCL) among patients with HS compared with patients without HS, being respectively 0,40% vs. 0,35%, 0,17% vs. 0,09%, and 0.06% vs. 0.02%. They stated that, because of the chronic inflammatory state in HS, the development of clonal immune cell populations may result in arising of malignant lymphomas; indeed, patients with HS appeared to have 2- to 4-times the risk of developing lymphomas compared with the general population. However, a limitation of their study was represented by the fact that they did not assess the disease duration and its severity [16]. In the setting of hematological conditions, Calamaro et al. also reported a case of intralymphatic proliferation of T-cell lymphoid blasts (IPTCLB) arising in a young male with HS of the inguinal region. IPTCLB is a benign condition characterized by the proliferation of large, atypical lymphoid cells (blastoid T lymphocytes expressing CD30) within lymphatic vessels, mimicking an intravascular lymphoma. The precise mechanisms at the basis of IPTCLB are unknown, as well as its possible link with HS [49]. In addition, a retrospective case-control study on 2292 patients (1776 with a validated diagnosis of HS) by Shlyankevich et al. highlighted the high comorbidity burden of patients with HS compared with that of matched control subjects. About 2% of the HS population (32/1730) also carried a diagnosis of lymphoma, whereas only 0.5% (9/1,730) of the controls had such diagnosis. The association of this finding is likely worth exploring given that several of the chronic inflammatory diseases (psoriasis, psoriatic arthritis, rheumatoid arthritis, Sjogren syndrome, IBD) are similarly associated with lymphoma. Increased immune activation and turnover of inflammatory cells may be a potential explanation for higher lymphoma rates in this population [12].

## 5. Conclusions

According to the pertinent literature, HS seems to be associated with an increased risk of several types of cancer. Some of these, such as NMSC, and particularly, SCC, represent a typical model of proliferative disease occurring on chronically injured site. Other types of malignancies, including hematological, oropharyngeal, CNS, colorectal, and prostatic ones were also found to be associated, the risk of which shows a direct correlation with HS severity. Our review suggests that cautious surveillance, through an age-appropriate cancer screening, and active intervention may be warranted in patients with HS, especially those with long-lasting and late-onset forms

## Figures and Tables

**Figure 1 biomedicines-09-01594-f001:**
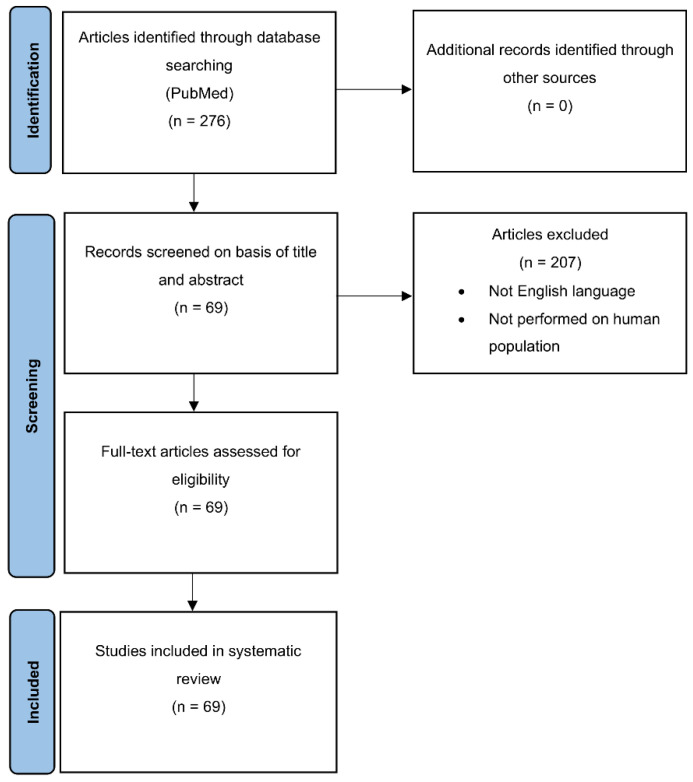
Flow diagram of studies assessed in systematic review. Diagram shows several step of studies selection in systematic review, also describing exclusion criteria and final number of included articles for each step.

**Table 1 biomedicines-09-01594-t001:** Strength levels assigned to different types of study in evaluation of reviewed articles (1 = highest, 6 = lowest).

Type of Article	Strength Level
Review	1
Retrospective cohort study	2
Case-control study	3
Cross sectional study	4
Case series	5
Case report	6

**Table 2 biomedicines-09-01594-t002:** Papers on correlation between Hidradenitis suppurativa (HS) and cancer. Strength level of each article is defined according to parameters shown in Table 1.

Authors, Reference Number and Year	Type of Study	Study Population	Strength Level
Gordon SW [17], 1977	Case report	38-year-old woman with HS and SCC of the sacral area	6
Black SB et al. [18], 1982	Case report	44-year-old man with HS and perianal SCC	6
Rosen T [19], 1986	Case report	55-year-old man with HS and perianal SCC	6
Zachary LS et al. [20], 1987	Case report	55-year-old man with HS and perineal SCC	6
Anstey AV et al. [21], 1990	Case report	67-year-old man with HS and SCC of the buttocks	6
Mendonça H et al. [22], 1991	Case report	57-year-old man with HS and SCC of the buttocks	6
Shukla VK et al. [23], 1995	Case report	71-year-old woman with HS and SCC of the buttocks	6
Pérez-Diaz D et al. [24], 1995	Review	27 cases described in literature presenting with HS and SCC. The buttocks were involved in 12 cases, perianal region in 11 cases, perineum in only one. The site of carcinoma was not specified in 3 cases. Four patients had lymph node metastasis, 1 lymph node, and liver metastatic disease	1
Dufresne RG Jr et al. [25], 1996	Case report	52-year-old woman with HS and follicular occlusion triad developing SCC of the buttocks	6
Malaguarnera M et al. [26], 1996	Case report	66-year-old man with perineal HS and metastatic SCC	6
Gur E et al. [27], 1997	Case report	1 case out of 4 with perineal SCC having also HS	6
Li M et al. [28], 1997	Case report	68-year-old man with HS and DD syndrome having perianal SCC	6
Ritz JP et al. [15], 1998	Retrospective cohort study	61-year-old man with gluteal SCC out of 31 patients that underwent surgery for HS	2
Manolitsas T et al. [29], 1999	Case report	52-year-old woman with HS and vulvar SCC	6
Cosman BC et al. [30], 2000	Case report	47-year-old man with HS and verrucous carcinoma (SCC) of the perianal area	6
Lapins J et al. [9], 2001	Retrospective cohort study	81 cases out of 2119 patients with HS. Significantly elevated relative risks for NMSC (5 cases), buccal cancer (5 cases), and primary liver cancer (3 cases)	2
Altunay IK et al. [31], 2002	Case report	54-year-old man with HS and SCC of the buttocks	6
Rosenzweig LB et al. [32], 2005	Case report	50-year-old man with HS and SCC of the perineum, complicated by paraneoplastic neuropathy	6
Short KA et al. [33], 2005	Case report	57-year-old woman with HS and SCC of the vulva	6
Nyquist GG et al. [34], 2007	Case series	31-year-old woman and 24-year-old man with HS and SCC of the groin in KID syndrome	5
Kurokawa I et al. [35], 2007	Case series	50-year-old man and 72-year-old man having HS and SCC of the buttocks, one with metastatic disease	5
Maclean GM et al. [36], 2007	Review	Description of 3 fatal cases of SCC in HS patients and review of other cases in literature: 15 patients had SCC of the buttocks, 9 had the involvement of the perianal region, 4 of the perineum, 2 of the vulva and 1 had SCC of the groin region. 13 of these presented lymph node metastasis	1
Barresi V et al. [37], 2008	Case report	47-year-old man with sacral HS and diffuse malignant peritoneal mesothelioma	6
Constantinou C et al. [38], 2008	Case series	One patient having diffuse abdominal carcinomatosis from perianal SCC arising on HS lesions; one patient with malignant hypercalcemia, as a complication of cutaneous SCC	5
Katz R [39], 2009	Case report	61-year-old man with HS and SCC (Marjolin Ulcer) of the buttocks	6
Chandramohan K et al. [40], 2009	Case report	40-year-old man with HS and perianal SCC	6
Lavogiez C et al. [41], 2010	Review	13 patients having HS and SCC of the buttocks, 2 with lymph node metastasis	1
Grewal NS et al. [42], 2010	Case series	2 cases with HS and gluteal SCC, 1 case with HS and SCC of the perineum	5
Scheinfeld N [43], 2014	Case report	47-year-old man with HS and perianal/anal SCC after starting infliximab	6
Chang JB et al. [44], 2014	Case report	50-year-old man with HS of the chest, axilla, thigh, buttocks, and groin and metastatic SCC (paraneoplastic syndrome: pancreatitis and deep venous thrombosis of the inferior vena cava and bilateral femoral veins)	6
Herschel S et al. [45], 2014	Case report	72-year-old man with HS and SCC of the sacral region with lymph nodes metastasis	6
Shlyankevich J et al. [12], 2014	Case control study	32 patients having HS and Lymphoma (9 healthy control), 9 patients having HS and SCC (0 healthy control)	3
Pena Z et al. [46], 2015	Case report	64-year-old woman having HS and metastatic vulvar SCC	6
Verdelli A et al. [47], 2016	Case report	78-year-old man with HS, treated with infliximab, and associated SCC	6
Rekawek P et al. [48], 2016	Case report	61-year-old woman with HS and SCC of the vulva	6
Calamaro P et al. [49], 2016	Case report	35-year-old man with inguinal HS and intralymphatic proliferation of T-cell lymphoid blasts	6
Joglekar K et al. [50], 2016	Case report	46-year-old man with gluteal HS and metastatic SCC	6
Makris GM et al. [3], 2017	Review	9 patients with HS and vulvar SCC (3 cases had nodal or metastatic disease); 6 patients with perineal/perianal SCC (2 cases had nodal or metastatic disease) and 1 with perianal mucinous adenocarcinoma	1
Huang C et al. [51], 2017	Case report	60-year-old man having HS and verrucous carcinoma (SCC) of the buttocks	6
Hessam S et al. [52], 2017	Case report	63-year-old man with inguinal, gluteal, and perianal HS and lymph node metastasis	6
Yon JR et al. [53], 2017	Review	2 cases out of 72 patients with HS, both having perineal SCC with lymph nodes metastasis	1
Zhang L et al. [54], 2017	Case report	59-year-old man with HS and SCC of the buttocks	6
Fabbrocini G et al. [55], 2017	Review	Over 90 cases in literature of HS and SCC developing on the perineal or buttock area	1
Sotoodian B et al. [56], 2017	Case series	a 60-year-old man with hairy cell leukemia and a 68 year-old man with chronic lymphocytic leukemia who developed HS later in life	5
Jourabchi N et al. [57], 2017	Case report	69-year-old man with HS and SCC of the buttocks	6
Alkeraye S et al. [28], 2017	Case report	61-year-old man with HS and perianal mucinous adenocarcinoma	6
McArdle DJT et al. [58], 2017	Case report	49-year-old woman with HS and anal SCC (carcinoma cuniculatum)	6
Dessinioti C et al. [59], 2017	Case report	56-year-old man with HS and SCC of the axillae	6
Chapman S et al. [4], 2018	Review	85 cases of SCC arising from HS found in English literature	1
Bessaleli E et al. [60], 2018	Case report	33-year-old woman treated with adalimumab who developed carcinoma of the cervix	6
Yen CF et al. [61], 2018	Case report	45-year-old man with HS and SCC of the buttock with metastatic disease	6
Beard C et al. [62], 2019	Case report	50-year-old man with MU arising in HS, having lymph node metastasis	6
Rastogi S et al. [14], 2019	Case control study	3 cases out of 716 patients with HS having vulvar SCC	3
Kohorst JJ et al. [13], 2019	Retrospective cohort study	12 cases with HS and gluteal, perianal, perineal and vulvo-vaginal SCC, 1 with lymph node metastasis	2
Tannenbaum R et al. [16], 2019	Cross sectional cohort study	Prevalence of NHL, HL, and CTCL among patient with HS (*n* = 62 690) compared with patient without HS (*n* = 20 937 880)	4
Nielsen VW et al. [63], 2020	Case report	66-year-old woman having HS, vulvar SCC, and metastatic disease	6
Ward R et al. [64], 2020	Case report	54-year-old woman with HS and SCC of the vulva	6

## Data Availability

Not applicable.

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
