# Peer review of "Neoplastic Implications in Patients Suffering from Hidradenitis Suppurativa under Systemic Treatments"

_biomedicines, 2021, doi:10.3390/biomedicines9111594_

Round 1
Reviewer 1 Report
I found the paper well-designed and well-prepared. However, in my opinion the Authors should comment on the assessment of risk of bias of selected papers for their final analysis. This will definitely add value to the manuscript. I also would like to know if this project was registered in PROSPERO database. If not, I will expect comment by the Authors.
Author Response
Dear Referee,
thank you for your useful suggestions, aimed to improve the quality of our paper. Please find enclosed, as a MS word file attached, the revised version of the paper.
We made several changes to the text, according with criticisms, listed below:
We reformatted the whole text that is now divided in three paragraphs within the discussion section. We hope that text in this “new” version should flow and result more comfortable for the readers. References have been also reformatted.
With regard to the unclear statements (comment #4):
- The data on the increased risk of NMSC from anti-TNF use come from psoriasis patients and includes study populations treated with conventional first line systemic agents and phototherapy. The statements included in the text are now clearly referred to the specific references;
- The discussion has been reformatted, with the aim to better explain the contents. Some paragraphs (e.g., the one on sunitinib have been also edited or omitted);
- The conclusion has been shortened. As it represents a summary of authors’ thinking on the topic, no references have been added.
Minor criticisms
- Abstract: the sentence has been reformatted with the aim of a better explanation of the content;
- Figure 1: the text bubbles are now aligned and easy to be read;
- The text has been checked for abbreviations and edited;
- Typo errors have been corrected.
Hoping that our manuscript, in its actual presentation, could be considered of interest for the readers of the journal, we thank you again for support and we send you our warmest regards.
Sincerely,
Laura Macca, M.D.
Reviewer 2 Report
Dear authors, the study is of great relevance and the methodology is sound, but I have some comments:
- Please insert references throughout the text, not just at the end. It is sometimes unclear if a statement refers to the previous reference or the last when there is a lot of text between references.
- Whole headings consist of long, multiple page paragraphs. E.g. under the heading: HS and Tu beyond the skin there is one reference in the beginning and one at the end of a long paragraph with multiple statements. Does the whole heading only discuss 2 papers? Are all statements taken only from single papers?
- All subheadings consist of single paragraphs. This is an unusual way to present. The heading 'HS and NMSC' goes over 4 pages. The text does not flow as change of subject is done many times within the same paragraph/heading. The paper is more difficult to read and separate subjects are difficult to locate within the paper. At the same time, references are scarce (see my previous comment).
On the other hand, you start a new heading (HS and Metastasis) with 'Although those SCC are...' Which SCC? You start a new heading and should explain which SCC you mean.
4. Unclear statement:
- On the use of TNF inh and increased risk of NMSC: you mention a two-fold risk and shorter time to developing NMSC particularly in psoriasis patients. Please clarify: Is the risk only evaluated in psoriasis patients or is there specific data on the risk in HS patients. Is this an extrapolated statement or a definite one?
Psoriasis patients would usually also have a history of phototherapies, other immunosuppressants (cyclosporine) and a longer history of TNF use compared to HS. Please clarify what is specific for HS, and what is only a speculation.
- In the discussion, it gets very confusing which statement refers to what. You mention the use of sunitinib to worsen HS. But you give no further information, eg was sunitinib used to treat a specific cancer in an already existing HS or a cancer not related to the HS? In the latter you want to point out the risk of treatment for other cancers that can worsen the HS. This is not new, and not specific for HS. Acne is a known side effect of EGFR inh. The point you want to make is unclear.
-The conclusion is too long and there are no references to support the claims.
Minor:
In the abstract you write 'disease-modifying comorbidities'. What do you mean?
Figure 1: Alignment in text bubbles should be corrected to make all text visible.
Page 5, line 218: You write T1, but nowhere in the text or under abbreviations is this explained.
Typo: Page 2, line 51: 'followe by'
Author Response
Dear Referee,
thank you for your useful suggestions, aimed to improve the quality of our paper. Please find enclosed, as a MS word file attached, the revised version of the paper.
We made several changes to the text, according with criticisms, listed below
A paragraph concerning the assessment of risk of bias has been added in the section “Materials and Methods”.
- Our study has not been registered in PROSPERO database as it was not expressly requested by journal instructions for authors
Hoping that our manuscript, in its actual presentation, should be considered of interest for the readers of the journal, we thank you for support and we send you our warmest regards.
Sincerely,
Laura Macca, M.D.
Round 2
Reviewer 2 Report
Dear authors, I cannot really see that my points are addressed.
Please provide a point by point answer to the comments and describe the changes that are made, as well as highlight changes in the text.
And at first glance, I also see that the alignment in the text bubbles is still not readable.
Author Response
Dear Reviewers,
thank you for your useful suggestions, aimed to improve the quality of our paper.
Please find enclosed, as a MS word file attached, the revised version of the paper. We used MS Word Track Changes software.
Several changes to the text have been made, according with criticisms, and are listed below:
Reply to referee #1
- We included references throughout the text;
- With regard to the paragraph on “HS and tumors beyond the skin”, it includes a very limited number of cases and papers actually present in literature. In facts, the two manuscript we cited represents the unique (and largest) studies in the field. Because of this reason, this part is really brief and constitutes the shortest chapter of our paper;
- We reformatted the whole text that is now divided in three paragraphs within the discussion section. We hope that text in this “new” version should flow and result more comfortable for the readers. References have been also reformatted.
The heading ‘HS and metastasis’ has been omitted. The contents of the original section have been included in the chapter ‘HS and NMSC’.
- The data on the increased risk of NMSC from anti-TNF use comes from psoriasis patients and includes study populations treated with conventional first line systemic agents and phototherapy. The statements included in the text are now clearly referred to the specific references;
- The discussion has been reformatted, with the aim to better explain the contents. Some paragraphs (e.g., the one on sunitinib have been also edited or omitted);
- The conclusion has been shortened. As it represents a summary of authors’ thinking on the topic, no references have been added.
Minor criticisms
- Abstract: the sentence has been reformatted with the aim of a better explanation of the content;
- Figure 1: the text bubbles are now aligned and easy to be read;
- The text has been checked for abbreviations and edited;
- Typo errors have been corrected.
Finally, the alignment in the text bubbles is now corrected and readable.
Reply to referee #2
- A paragraph concerning the assessment of risk of bias has been added in the section “Materials and Methods”.
- Our study has not been registered in PROSPERO database as it was not expressly requested by journal instructions for authors.
Hoping to have adequately responded to the referees’ suggestions, and that the manuscript in its actual presentation could be considered by you for the readers of the journal, we send you all the best regards,
Sincerely yours,
Claudio Guarneri, M.D. (on behalf of the coauthors)